# The value of D-dimer and platelet-lymphocyte ratio combined with CT signs for predicting intestinal ischemia in patients with bowel obstruction

Yuan Zhou[1,2], Haijian Zhao[2], Bing Liu[1,2], Jiangfeng Qian[1,2], Ning Chen[2], Yan Wang[2], Daoyuan Tu[2], Xiaoyu Chen[3], Heng Li[4], Xiaoyu Zhang🔘[2]*

1 Xuzhou Medical University, Xuzhou, China, 2 Department of Gastrointestinal Surgery, The Affiliated Huai'an Hospital of Xuzhou Medical University, Huai'an, China, 3 Department of Radiology, The Affiliated Huai'an Hospital of Xuzhou Medical University, Huai'an, China, 4 Department of Pathology, The Affiliated Huai'an Hospital of Xuzhou Medical University, Huai'an, China

* yllzxy@163.com

**Data Availability Statement:** All relevant data are within the manuscript and its Supporting Information files.

## Abstract

### Objective

To investigate the diagnostic value of D-dimer, platelet-lymphocyte rate (PLR) and CT signs for intestinal ischemia in patients with bowel obstruction.

### Methods

We retrospectively analyzed the clinical and imaging data of 105 patients diagnosed with bowel obstruction, and performed univariate and multivariate analyses to determine the independent risk factors for intestinal ischemia in patients with bowel obstruction. Moreover, the receiver operating characteristic curve (ROC) was plotted to examine the diagnostic value of D-dimer, PLR and CT signs in patients with bowel obstruction. Besides, Kappa tests were used to assess inter-observer agreement.

### Results

We included 56 men (53%) and 49 women (47%) with mean age of 66.05 ± 16 years. Univariate and multivariate analyses showed that D-dimer, PLR and two significant CT signs (i.e., increased unenhanced bowel-wall attenuation and mesenteric haziness) were independent risk factors for intestinal ischemia in patients with bowel obstruction. ROC analysis showed that the combined use of D-dimer, PLR and the said two CT signs had better performance than single indicators in predicting intestinal ischemia in patients with bowel obstruction. The area under the curve (AUC) of the joint model III was 0.925 [95%CI: 0.876–0.975], with a sensitivity of 79.2% [95CI%: 67.2–91.1] and a specificity of 91.2% [95%CI: 83.7–98.9].

**Funding:** This work was supported by grants from Natural Science Foundation of Huai'an Municipality and Training Program of "Six Talent Peaks" in Jiangsu Province to Xiaoyu Zhang [grant number: HAB202212; WSW291 respectively].

## Conclusion

The combined use of D-dimer, PLR and CT signs has high diagnostic value for intestinal ischemia in patients with bowel obstruction and will prompt surgical exploration to evaluate intestinal blood flow.

## 1. Introduction

Bowel obstruction is the third most common acute abdominal disease in abdominal surgery after acute appendicitis and biliary tract diseases. When an obstruction arises in the intestinal tract or the movement of intestinal contents is obstructed, it can lead to systemic physiological disorders and anatomical and functional alterations in intestinal tubes. Such complications can be life-threatening in severe cases. Even if the bowel obstruction is resolved at a later stage, the patient may still be susceptible to death due to significant pathophysiologic and pathoanatomic changes. Regarding the management of acute bowel obstruction, international guidelines always advocate conservative management if there is no clinical and/or radiologic evidence of strangulation and intestinal ischemia [1]. The conservative therapy has achieved satisfactory results in approximately 80% of bowel obstruction cases [2], but it leads to a mortality rate of 10–35% when blood supply disorders occur [3–5]. Therefore, early recognition of intestinal ischemia is critical for making surgical decisions, presenting a great challenge for clinicians. Delayed diagnosis and surgery may cause intestinal perforation, infectious shock, and even death of in patients [6, 7] Rapid and aggressive surgical interventions may exacerbate the condition of patients, add their financial load, and increase the risk of subsequent adhesive bowel obstruction, resulting in bowel obstruction recurrence [8].

D-dimer, a product of fibrin degradation, is widely accepted as the most reliable laboratory marker for detecting coagulation activation. It is mainly clinically applied to the elimination of the possibility of thrombotic disorders, including deep vein thrombosis or pulmonary embolism. Although D-dimer shows high sensitivity in the diagnosis of acute intestinal ischemia, it is less commonly used in clinical practice because of its lower specificity [9].

Platelet-lymphocyte ratio (PLR) is mainly used as a biomarker of the systemic inflammatory response and shows a strong correlation with immune diseases, vascular diseases and certain tumors. Several recent studies have demonstrated the potential of PLR for predicting the prognosis of patients with mesenteric ischemia. Additionally, it has been proven that PLR has potential applications in the diagnosis of mesenteric ischemia, with a sensitivity of 59% and a specificity of 65% [10, 11]. However, PLR alone showed poor diagnostic effect and instable detection effect.

Currently, computed tomography (CT) is a preferred imaging technique for assessing the patient's condition and identifying the location and cause of obstruction. Numerous studies have demonstrated that CT signs are strongly linked to intestinal ischemia [12]. In most studies, the sensitivity and specificity of CT in diagnosing bowel-wall ischemia in patients with small bowel obstruction (SBO) ranged from 73% to 100% and from 61% to 93%, respectively [13]. However, a recent prospective multicenter study has revealed that CT has a sensitivity of only 40% in diagnosing ischemia-related complications in patients with bowel obstruction [14].

The purpose of our study is to evaluate the performance and accuracy of D-dimer, PLR and CT signs for the diagnosis of bowel-wall ischemia in patients with bowel obstruction based on surgical and histopathologic findings.

## 2. Materials and methods

### 2.1 Study population

All consecutive patients admitted to the gastrointestinal surgery department of our institution from January 2020 to July 2023 for bowel obstruction were retrospectively reviewed. Among the 135 patients with bowel obstruction in this study, 30 were initially excluded. The reasons for exclusion are detailed in the study flowchart (Fig 1). In total, 105 patients met the inclusion criteria in this study, including 56 males and 49 females. The group was divided into two categories based on surgical and pathological findings: the intestinal ischemia group (48 cases) and the non-ischemia group (57 cases).

### 2.2 Methods

Our hospital review board approved this study and informed consent for the use of medical records was obtained from patients. Data regarding patient demographics, imaging findings, blood test results and findings at the time of operation were collected (from December 1,2022 to August 31, 2023). All data were retrospectively retrieved from the electronic medical record system.

Prior to admission or surgery, all patients were examined by CT scan using a Aquilion one (Toshiba, Japan)) from bilateral diaphragmatic domes to symphysis pubes. Scan parameters were set as below: tube voltage: 120 kV, tube current: 250 mA, slice thickness: 1 mm.

### 2.3 Reference standard

Each patient's medical history was obtained retrospectively, and the patients were divided into two groups based on surgical and pathological report data. φ Non-ischemia group: intraoperative observation revealed reddish bowel color and normal peristalsis, and only the obstruction

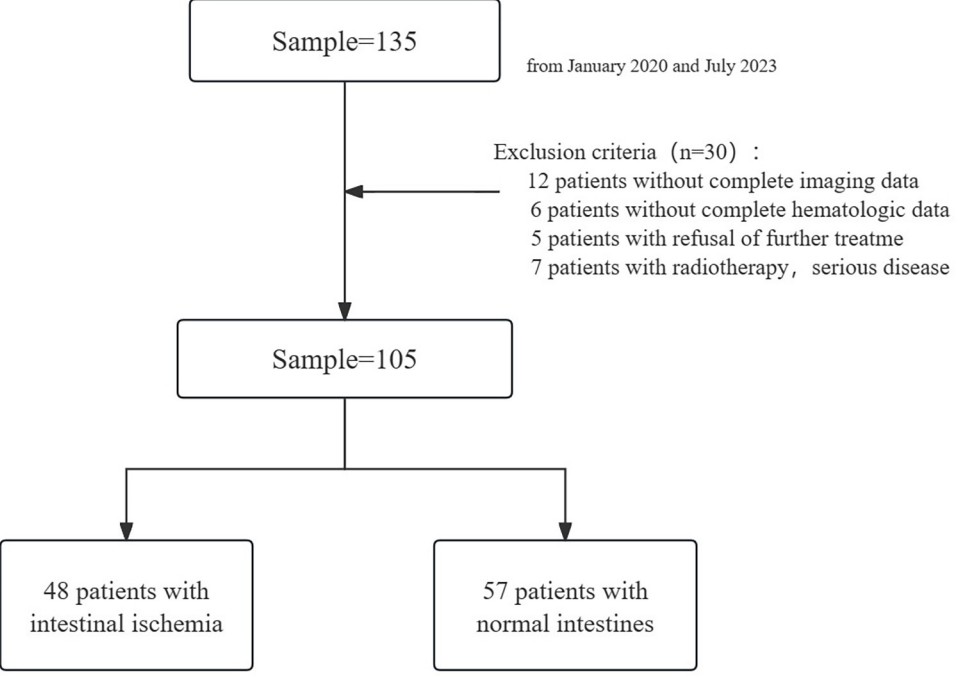

**Fig 1. Flowchart of patient enrollment process.**

was relieved intraoperatively without resection of the bowel segments; κ Intestinal ischemia group: including reversible and irreversible intestinal ischemia. During the operation, it was observed that the color of the bowel-wall had changed and peristalsis had weakened or ceased, following the release of the obstruction, the affected intestinal segment was placed on a warm gauze soaked in saline for 15–20 minutes for observation. If the intestinal segment regains its normal color and peristalsis, and the obstruction is released intraoperatively without resection of the intestinal segment, then it is classified as reversible intestinal ischemia. If the viability of the obstructed bowel segment is not restored, the bowel-wall is purplish in color, peristalsis is absent, the mesenteric arteries are not pulsatile requiring segmental resection, and the pathology report specifies intestinal necrosis, then it is classified as irreversible intestinal ischemia.

## 2.4 Image analysis

Two radiologists (both with 5 or more years of experience) retrospectively and independently reviewed all CTs, and disagreements were resolved by a third reader to reach final consensus. Image analysis is performed at the Picture Archiving and Communication Systems (PACS) terminal, combining axial and coronal images, with the ability to arbitrarily adjust the image window width and position, and to complete the assessment content independently and blindly without knowledge of the surgical and pathological findings. Subjective assessment of signs included the degree of bowel-wall enhancement, bowel-wall thickness, peritoneal effusion, mesenteric ambiguity, and the whirlpool sign; each sign was analyzed comprehensively for the presence of each sign and the degree of its presence.

The CT signs were analyzed based on the following parameters: (1) Increased unenhanced bowel-wall attenuation defined as high density of the bowel wall of a dilated loop compared to a healthy dilated loop on unenhanced CT images; (2) Mesenteric haziness defined as increased mesenteric fat attenuation; (3) Bowel wall thickening defined as bowel wall thickness greater than 5 mm; (4) Peritoneal fluid defined as fluid within the peritoneal cavity, excluding mesenteric fluid; (5) Whirl sign, defined as a swirled appearance of the mesenteric fat and vessels at the root of the mesentery with an adjacent rotated bowel loop. Fig 2 showed examples of the signs.

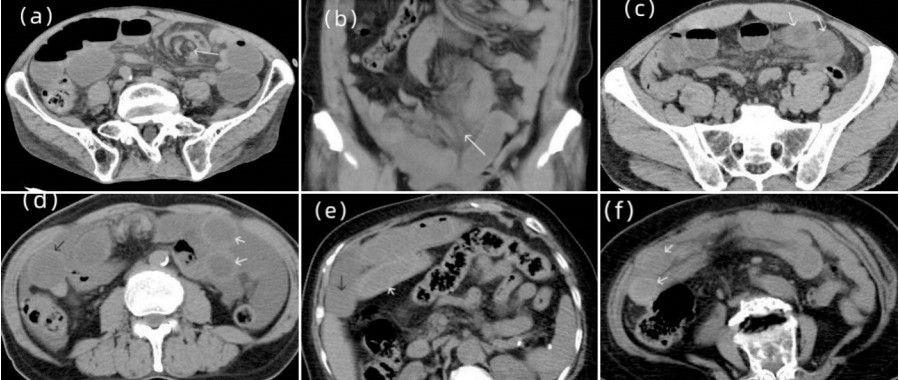

**Fig 2.** The signs are as follows: (a) Axial abdominal CT scan shows whirl sign (white arrow) (b) Corona abdominal CT scan shows mesenteric haziness (white arrow) (c) Axial abdominal CT scan shows bowel wall thickening (white arrows) (d) Axial abdominal CT scan show increased unenhanced bowel-wall attenuation (white arrows), peritoneal fluid seen around the bowel wall (e) Axial abdominal CT scan shows increased density of the bowel wall of dilated loop (white arrow) compared to surrounding healthy bowel (black arrow) (f) Axial abdominal CT scan shows bowel wall thickening (white arrows).

## 2.5 Statistical analysis

Comparison of all clinical variables and CT findings in the non-ischemia group and the intestinal ischemia group. Descriptive statistics were used for baseline characteristic, with continuous variables expressed as mean and standard deviation (SD) if the distribution was normal, or as median and interquartile range (IQR) if the distribution was not normal, and categorical variables as number and percentage. For normally distributed variables, t-tests were used for comparisons of quantitative variables, otherwise Mann-Whitney U-tests were used. Comparisons between groups for categorical variables were made using the Pearson chi-square test or Fisher exact test. When Youden index (sensitivity + specificity -1) is the largest, the cut-off value of continuous variables is the boundary value.

In univariate analyses, all variables associated with p-value <0.05 were entered into a multivariate logistic regression model. When statistically significant variables were identified, sensitivity, specificity, Positive predictive value (PPV), and Negative predictive value (NPV) were estimated using subject operating characteristic curve (ROC) based analysis. Inter-observer agreement was determined by the kappa test. Kappa values ranging 0.00–0.20, 0.21–0.40, 0.41–0.60, 0.61–0.80, and 0.81–1.00 indicated slight, fair, moderate, substantial, and almost perfect agreement, respectively. All statistical analyses were performed using SPSS software (SPSS, version 26.0).

# 3. Results

## 3.1 Population characteristics

A total of 105 patients (consisting of 56 men (53%) and 49 women (47%)) were included in the study, with mean age of 66.05 ± 16 years old. There was no statistically significant difference between the two groups in age, duration of pain, gender, location of obstruction, history of abdominal surgery, neutrophil level, platelet level, and albumin level (all P>0.05). Compared with the non-ischemia group, the group with intestinal ischemia presented more significant peritoneal irritation signs and CT signs (increased unenhanced bowel-wall attenuation, bowel-wall thickening, mesenteric haziness, peritoneal fluid, and whirl sign), a higher D-dimer level, a higher neutrophil-lymphocyte ratio (NLR), a higher PLR, and a higher C-reactive protein level (all P<0.05) (Table 1).

## 3.2 CT signs

According to Table 2, the inter-reader agreement was substantial to almost perfect for all CT features. The agreement for the whirl sign was substantial (k = 0.767, 95%CI: [0.676–0.858]), and for increased unenhanced bowel-wall attenuation, bowel-wall thickening, mesenteric haziness, and peritoneal fluid was almost perfect (k = 0.895, 95%CI: [0.844–0.946]; k = 0.886, 95%CI: [0.822–0.950]; k = 0.847, 95%CI: [0.795–0.899]; k = 0.882, 95%CI: [0.835–0.929], respectively).

## 3.3 Analysis results of variables

The multivariate analysis showed that D-dimer (OR = 2.450, 95%CI: [0.162–65.359], p = 0.046), PLR (OR = 1.010, 95%CI: [1.000–1.020], p = 0.044), increased unenhanced bowel-wall attenuation (OR = 20.003, 95%CI: [1.500–266.745], p = 0.023) and mesenteric haziness (OR = 10.841, 95%CI: [2.329–50.452], p = 0.002) were independent risk factors for intestinal ischemia in patients with bowel obstruction (Table 3).

**Table 1. Clinical data and CT signs.**

| Variables | Non-ischemia group (n = 57) | Intestinal ischemia group (n = 48) | $\chi^2/t/Z$ | P value |
|---|---|---|---|---|
| Age, n (%) | | | 1.696 | 0.193 |
| < = 60 | 21 (36.8%) | 12 (25%) | | |
| >60 | 36 (63.2%) | 36 (75%) | | |
| Duration of pain [Q(IQR), hour] | 48.00[12.5,72] | 48.00[12,96] | -0.582 | 0.560 |
| Sex, n (%) | | | 0.056 | 0.814 |
| Male | 31 (54.4%) | 25 (52.1%) | | |
| Female | 26 (45.6%) | 23 (47.9%) | | |
| Position, n (%) | | | 0.395 | 0.529 |
| Small intestine | 51 (89.5%) | 41 (85.4%) | | |
| Large intestine | 6 (10.5%) | 7 (14.6%) | | |
| History of abdominal surgery (Yes), n (%) | 42 (73.7%) | 29(60.4%) | 2.095 | 0.148 |
| Peritoneal irritation sign (Yes), n (%) | 2 (3.5%) | 13 (27.1%) | 11.827 | 0.001 |
| Laboratory indicators | | | | |
| D-dimer, [Q(IQR)] | 0.67[0.41,1.10] | 1.43[0.89,2.23] | | <0.001 |
| Neutrophil count, [Q(IQR),10*9/L] | 7.83[5.14,10.16] | 5.09[4.15,9.26] | -1.640 | 0.101 |
| Lymphocyte count [Q(IQR),10*9/L] | 1.08[0.75,1.56] | 0.73[0.50,0.93] | -4.481 | <0.001 |
| Neutrophil-lymphocyte ratio, [Q(IQR), 10*9/L] | 6.49[4.08,10.82] | 8.26[5.48,12.72] | -2.148 | 0.032 |
| Platelet count, [Q(IQR),10*9/L] | 208[180.50,238.00] | 210.5[137.75,262.25] | -0.029 | 0.977 |
| Platelet-lymphocyte ratio, [Q(IQR),10*9/L] | 192.59[146.05,265.27] | 287.13[208.00,390.38] | -4.451 | <0.001 |
| Albumin, (mean± SD) | 41.36±6.93 | 40.63±8.52 | 0.489 | 0.626 |
| C-reactive protein, [Q(IQR)] | 3.55[0.88,23.41] | 13.15[3.48,60.60] | -2.294 | 0.003 |
| CT signs, n (%) | | | | |
| Increased unenhanced bowel-wall attenuation | 2(3.5%) | 22(45.8%) | 26.472 | <0.001 |
| Bowel wall thickening | 1(1.8%) | 13(27.1%) | 19.114 | <0.001 |
| Mesenteric haziness | 19(33.3%) | 38(79.2%) | 22.057 | <0.001 |
| Peritoneal fluid | 28(49.1%) | 35(72.9%) | 6.147 | 0.013 |
| Whirl sign | 3(5.3%) | 10(20.8%) | 5.823 | 0.019 |

The data are number (%)

## 3.4 Diagnostic performance of indicators and joint models

In this study, the combinations of D-dimer + PLR + increased unenhanced bowel-wall attenuation, D-dimer + PLR + mesenteric haziness, and D-dimer + PLR + increased unenhanced bowel-wall attenuation + mesenteric haziness were defined as joint models I, II, and III, respectively. Diagnostic values (i.e., sensitivity, specificity, and AUC) are shown in Table 4. The results showed that joint models had better diagnostic performance than single indicators

**Table 2. Inter-reader agreement for CT signs.**

| N = 105 | Reader 1 | Reader 2 | Agreement | % | Kappa | 95%CI |
|---|---|---|---|---|---|---|
| Increased unenhanced bowel-wall attenuation | 24 | 26 | 101/105 | 96% | 0.895 | 0.844–0.946 |
| Bowel wall thickening | 15 | 16 | 102/105 | 97% | 0.886 | 0.822–0.950 |
| Mesenteric haziness | 56 | 58 | 97/105 | 92% | 0.847 | 0.795–0.899 |
| Peritoneal fluid | 61 | 63 | 99/105 | 94% | 0.882 | 0.835–0.929 |
| Whirl sign | 14 | 15 | 98/105 | 93% | 0.767 | 0.676–0.858 |

95% CI 95% confidence interval

**Table 3. The univariate and multivariate analysis of two groups.**

| Variables | Univariate analysis | | | Multivariate analysis | | |
|---|---|---|---|---|---|---|
| | OR | 95%CI | P value | OR | 95%CI | P value |
| Age (>60) | 1.75 | 0.751–4.08 | | | | |
| Duration of pain | 1.002 | 0.998–1.005 | 0.327 | | | |
| Sex (Female) | 0.912 | 0.422–1.968 | 0.814 | | | |
| Position (Small intestine) | 1.451 | 0.453–4.654 | 0.531 | | | |
| History of abdominal surgery (Yes) | 1.834 | 0.803–4.190 | 0.150 | | | |
| Peritoneal irritation sign (Yes) | 10.214 | 2.173–48.021 | 0.003 | 3.259 | 0.162–65.359 | 0.440 |
| Laboratory indicators | | | | | | |
| D-dimer | 4.171 | 2.027–8.581 | <0.001 | 2.450 | 1.017–5.903 | 0.046 |
| Neutrophil count | 1.101 | 0.911–1.099 | 0.543 | | | |
| Lymphocyte count | 0.109 | 0.034–0.35 | 0.001 | 0.480 | 0.064–3.610 | 0.476 |
| Neutrophil-lymphocyte ratio | 1.101 | 1.021–1.188 | 0.012 | 0.908 | 0.789–1.045 | 0.179 |
| Platelet count | 1.001 | 0.997–1.006 | 0.598 | | | |
| Platelet-lymphocyte ratio | 1.011 | 1.006–1.016 | <0.001 | 1.010 | 1.000–1.020 | 0.044 |
| Albumin | 0.987 | 0.938–1.039 | 0.662 | | | |
| C-reactive protein | 1.005 | 0.998–1.012 | 0.141 | | | |
| CT signs | | | | | | |
| Increased unenhanced bowel-wall attenuation | 23.269 | 5.085–106.481 | <0.001 | 20.003 | 1.500–266.745 | 0.023 |
| Bowel wall thickening | 20.8 | 2.265–166.049 | 0.004 | 1.706 | 0.109–26.597 | 0.703 |
| Mesenteric haziness | 7.6 | 3.127–18.47 | <0.001 | 10.841 | 2.329–50.452 | 0.002 |
| Peritoneal fluid | 2.788 | 1.226–6.341 | 0.014 | 1.055 | 0.257–4.327 | 0.940 |
| Whirl sign | 4.737 | 1.221–18.369 | 0.024 | 3.693 | 0.555–24.552 | 0.176 |

*CT* computed tomography; *OR* odds ratio; *CI* confidence interval

in all cases. Specifically, the AUC, sensitivity, specificity, PPV, and NPV of the joint model III was 0.925 [95%CI: 0.876–0.975], 79.2% [95%CI%: 67.2–91.1], 91.2% [95%CI: 83.7–98.9], 88.4% [95%CI: 78.4–98.4], and 83.9% [95%CI: 74.5–93.3], respectively. The diagnostic performance of the joint model III was better than that of other models. The single marker, increased unenhanced bowel-wall attenuation, and the joint model I had a higher specificity (96.5% [95%CI: 91.6–101.4]), but a poor sensitivity (45.8% [95%CI: 31.2–60.5]; 68.8% [95%CI: 55.1–

**Table 4. Summary performance value for each indicator for diagnosing ischemia.**

| | Sensitivity | Specificity | PPV | NPV | AUC | 95%CI |
|---|---|---|---|---|---|---|
| D-dimer | 75.0[62.3,87.7] | 66.7[54.0,79.3] | 65.5[52.5,78.4] | 76.0[63.7,88.3] | 0.766 | 0.647–0.859 |
| PLR | 70.8[57.5,84.2] | 70.2[57.9,82.4] | 66.7[53.3,80.1] | 74.1[62.0,86.1] | 0.753 | 0.661–0.845 |
| Increased unenhanced bowel-wall attenuation | 45.8[31.2,60.5] | 96.5[91.6,101.4] | 91.7[79.7,103.6] | 67.9[57.5,78.3] | 0.712 | 0.609–0.815 |
| Mesenteric haziness | 79.2[67.2,91.1] | 66.7[54.0,79.3] | 66.7[67.2,91.1] | 79.2[54.0,79.3] | 0.729 | 0.631–0.828 |
| Joint model I | 68.8[55.1,82.4] | 96.5[91.6,101.4] | 94.3[86.2,102.4] | 78.6[68.7,88.4] | 0.874 | 0.804–0.944 |
| Joint model II | 81.3[69.8,92.7] | 84.2[74.4,94.0] | 81.3[69.8,92.7] | 84.2[74.4,94.0] | 0.886 | 0.824–0.948 |
| Joint model III | 79.2[67.2,91.1] | 91.2[83.7,98.9] | 88.4[78.4,98.4] | 83.9[74.5,93.3] | 0.925 | 0.876–0.975 |

Data in brackets are 95% confidence intervals

*PPV* positive predictive value; *NPV* negative predictive value; *PLR* platelet-lymphocyte ratio

Joint model I was defined as D-dimer and PLR combined with increased unenhanced bowel-wall attenuation; Joint model II was defined as D-dimer and PLR combined with mesenteric haziness; Joint model III was defined as the union of the four.

82.4]) for diagnosing intestinal ischemia in patients with bowel obstruction. The sensitivity (81.3% [95%CI: 69.8–92.7]) and specificity (84.2% [95%CI: 74.4–94.0]) of the joint model II were high, with an AUC of 0.886 [95%CI: 0.824–0.948], indicating good diagnostic effect of this model.

## 4. Discussion

This retrospective study showed that D-dimer, PLR, increased unenhanced bowel-wall attenuation, and mesenteric haziness were significantly associated with intestinal ischemia in patients, and they helped identify patients with bowel obstruction complicated by intestinal ischemia. We established several joint models of these four factors, which showed good diagnostic performance. The sensitivities and specificities of the joint models constructed for diagnosing bowel-wall ischemia in patients with SBO ranged from 68.8% to 81.3% and from 84.2% to 96.5%, respectively.

The physiological mechanisms of intestinal ischemia and necrosis are primarily arterial or venous occlusion, or non-occlusive reduction of arterial or venous flow [15]. The causes for intestinal ischemia in patients with bowel obstruction are as follows. The first reason is vascular mechanical obstruction arising from the twisting of intestinal collaterals. The second reason is the dilatation of the obstructing collaterals caused by compression, which leads to microcirculatory obstruction of both arterial and venous systems and finally hypoxia. The third cause is venous congestion resulted from the dilation of collaterals [16].

Serum dextro-lactic acid, intestinal-type fatty acid binding protein, and circulating deoxyribonucleic acid are laboratory markers with relatively high specificity [17–19]. However, these markers are controversial as they are still in the early stages of research and have not yet been widely used in clinical settings. The NLR predicted intestinal ischemia with a sensitivity of 85.1% and a specificity of 63%, while the PLR had a poor predictive effect, as demonstrated in the study of Evangeline et al. [20]. The PLR also has potential in evaluating the prognosis of patients with acute mesenteric ischemia within 30 days, as suggested in the research by Emmanuel et al. [11]. Previous studies have shown that D-dimer is an effective marker of intestinal ischemia, with a sensitivity of 96% and a low specificity of approximately 40% [21]. It is evidenced that decreased bowel-wall enhancement in enhanced CT images is the most reliable indicator of intestinal ischemia [22, 23]. However, the enhanced CT examination requires intravenous injection of a contrast medium, so it is not used as a routine procedure for patients with bowel obstruction in many medical units in China. Increased unenhanced bowel-wall attenuation was significantly associated with intestinal ischemia, showing a specificity of 100% and a low sensitivity of 56% in the prediction of the disease, according to Geffroy et al. [16]. This result is broadly in line with the finding of the present study, which showed that increased unenhanced bowel-wall attenuation had a sensitivity of 96.5% and a specificity of 45.8% in predicting intestinal ischemia. In the present study, the intestinal hematological status of patients with bowel obstruction was comprehensively assessed from both hematological and imaging perspectives. The aim is to provide guidance for clinicians on the making of treatment decisions for patients with bowel obstruction, slow down the progression of the positive intestinal tube towards ischemic necrosis, perforation, and toxic shock, and thus to improve the quality of survival and prognosis of the patients.

Our study revealed that peritoneal irritation signs, D-dimer, lymphocyte values, PLR, NLR, and CT signs were strongly associated with intestinal ischemia in patients with bowel obstruction (all P<0.05). The multivariate analysis showed that D-dimer, PLR, increased unenhanced bowel-wall attenuation, and mesenteric haziness were independent risk factors for intestinal hemodynamic disorders in patients with bowel obstruction. In addition, according to the

evaluation results of the effect of the above four indicators for the diagnosis of intestinal ischemia, the AUC values of D-dimer, PLR, increased unenhanced bowel-wall attenuation, and mesenteric haziness were 0.766, 0.753, 0.712, and 0.729, respectively. Among them, increased unenhanced bowel-wall attenuation had the best specificity (96.5%) and a positive predictive value of 91.7%, which was better than that of PLR, D-dimer, and mesenteric haziness. However, it had a poor sensitivity (45.8%). Therefore, we combined these four indicators and constructed three joint models. The results suggested that the AUC values of joint models I, II, and III were 0.874, 0.886, and 0.925, respectively, which were better than those of single indicators. The diagnostic effect of the joint model III was the best. The joint model compensated for the shortcomings of poor sensitivity or specificity of single indices, improving the accuracy of diagnosing intestinal ischemia in clinical practice.

The main pathophysiological changes in bowel obstruction are oedema and ischemia, which eventually progress to intestinal perforation necrosis if no correct therapeutic decisions are made. On CT images, increased unenhanced bowel-wall attenuation is associated with submucosal hemorrhage of the bowel wall, which is caused by hemorrhagic necrosis within the intestinal wall and the extravasation of blood components via damaged capillary walls. In patients with strangulated bowel obstruction, the mesenteric blood vessels may become compressed or obstructed due to greater external pressure applied on the arteries than that on the veins. The obstruction of venous return can lead to persistent arterial blood flow and the accumulation of blood in the capillary system. As a result, both the intravascular hydrostatic pressure and the permeability of the micrangium wall increases, giving rise to plasma extravasation. If compression or obstruction persists, arterial blood flow is impeded, thereby bringing about hypoxia. This condition further damages the micrangium wall, promoting the extravasation of red blood cells into the intestinal wall and the formation of hemorrhagic foci. Mesenteric haziness is related to mesenteric blood supply [15, 24]. The present study demonstrated that increased unenhanced bowel-wall attenuation had a good specificity but a poor sensitivity for the prediction of intestinal ischemia. Its sensitivity increased to 79.2% when it was combined with other indicators. The study by Wiesner et al. [25] suggested that changed mesenteric density was a reliable sign of SBO strangulation. Our study results indicated that mesenteric haziness had a good sensitivity and specificity in predicting intestinal ischemia in patients with bowel obstruction, which is consistent with the conclusion drawn by Liu et al. [24] and Millet et al. [26].

Notably, our study is mainly based on surgical patients as the research object. The main reasons for this are as follows: For patients who have not undergone surgical treatment. There are two possible outcomes for those patients: (1) successful conservative treatment; (2) failed conservative treatment. Firstly, patients with successful conservative treatment have been cured and discharged from hospital and returned to normal life. The patient's intestinal blood flow was completely normal during the time of this admission. Secondly, there are also only two outcomes in patients with conservative treatment failure: surgical treatment and abandonment of treatment (non-cooperation with treatment). The former were included in this study, and patients who did not cooperate with the treatment met the exclusion criteria and were not included in the study. In conclusion, a total of emergency surgical patients and patients who failed conservative treatment and were operated were included in this study. Whether there is a difference between these two this needs to be further evaluated in large sample, multicenter studies for comparison.

Our study has several limitations. Firstly, it is a retrospective, single-centre study. Prospective, large-sample randomized controlled studies are needed to prove our conclusions in the future. Secondly, all patients only underwent conventional CT scans. Enhanced CT and dual-energy CT may provide more information about bowel-wall ischemia. Lastly, retrospective

selection might lead to inclusion bias, and most of the patients with clinically significant bowel obstruction were included.

## 5. Conclusion

In conclusion, the combination of D-dimer, PLR and CT signs can provide a more accurate and comprehensive assessment of the intestinal hemodynamic status of patients with bowel obstruction. The combined use of the indicators also has better performance in the prediction of intestinal ischemia. D-dimer, PLR, increased unenhanced bowel-wall attenuation, and mesenteric haziness are independent risk factors for intestinal ischemia in patients with bowel obstruction. In addition, the joint models have better diagnostic effect than single indicators, and they can provide more accurate, comprehensive and reliable information to guide clinical treatment.

## Supporting information

**S1 Data. Raw data set.**
(XLSX)

## Author Contributions

**Conceptualization:** Yuan Zhou, Haijian Zhao.

**Data curation:** Yuan Zhou, Daoyuan Tu.

**Formal analysis:** Jiangfeng Qian, Ning Chen, Yan Wang.

**Investigation:** Yuan Zhou, Bing Liu, Jiangfeng Qian, Ning Chen, Xiaoyu Chen, Heng Li.

**Methodology:** Yuan Zhou, Bing Liu.

**Writing – original draft:** Yuan Zhou.

**Writing – review & editing:** Haijian Zhao, Xiaoyu Zhang.

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
