## [Decision Letter · Decision Letter 0]

16 Apr 2024

PONE-D-24-07576The value of D-dimer and platelet-lymphocyte ratio combined with CT signs for predicting intestinal ischemia in patients with bowel obstructionPLOS ONE

Dear Dr. Zhang,

Thank you for submitting your manuscript to PLOS ONE. After careful consideration, we feel that it has merit but does not fully meet PLOS ONE’s publication criteria as it currently stands. Therefore, we invite you to submit a revised version of the manuscript that addresses the points raised during the review process.

We look forward to receiving your revised manuscript.

Kind regards,

Yoshihisa Tsuji

Academic Editor

PLOS ONE

Journal Requirements:

2. Please provide additional details regarding participant consent. As you are reporting a retrospective study of medical records or archived samples, please ensure that you have discussed whether all data were fully anonymized before you accessed them and/or whether the IRB or ethics committee waived the requirement for informed consent. If patients provided informed written consent to have data from their medical records used in research, please include this information.

   "This work was supported by grants from Natural Science Foundation of Huai’an Municipality and Training Program of “Six Talent Peaks” in Jiangsu Province to Xiaoyu Zhang [grant number: HAB202212; WSW291 respectively]"

4. In this instance it seems there may be acceptable restrictions in place that prevent the public sharing of your minimal data. However, in line with our goal of ensuring long-term data availability to all interested researchers, PLOS’ Data Policy states that authors cannot be the sole named individuals responsible for ensuring data access (http://journals.plos.org/plosone/s/data-availability#loc-acceptable-data-sharing-methods).

6. Please note that funding information should not appear in any section or other areas of your manuscript. We will only publish funding information present in the Funding Statement section of the online submission form. Please remove any funding-related text from the manuscript.

7. Your ethics statement should only appear in the Methods section of your manuscript. If your ethics statement is written in any section besides the Methods, please move it to the Methods section and delete it from any other section. Please ensure that your ethics statement is included in your manuscript, as the ethics statement entered into the online submission form will not be published alongside your manuscript. 

8. We note you have included a table to which you do not refer in the text of your manuscript. Please ensure that you refer to Table 3 in your text; if accepted, production will need this reference to link the reader to the Table.

Additional Editor Comments:

Although the study was interesting, it appeared that this version of the draft did not reach the publishable label. Especially, please describe the details of the entry criteria more clearly, as pointed out by reviewer number 2.

Reviewers' comments:

Reviewer's Responses to Questions

**Comments to the Author**

1. Is the manuscript technically sound, and do the data support the conclusions?

Reviewer #1: Yes

Reviewer #2: Yes

2. Has the statistical analysis been performed appropriately and rigorously? 

Reviewer #1: Yes

Reviewer #2: I Don't Know

3. Have the authors made all data underlying the findings in their manuscript fully available?

Reviewer #1: Yes

Reviewer #2: No

4. Is the manuscript presented in an intelligible fashion and written in standard English?

Reviewer #1: Yes

Reviewer #2: Yes

5. Review Comments to the Author

Reviewer #1: I have read this paper title” The value of D-dimer and platelet-lymphocyte ratio combined with CT signs for predicting intestinal ischemia in patients with bowel obstruction”. I find both the METHOD and RESULT very interesting and the discussion is logical.

I had some questions after reading this paper, but all of them are mentioned in the limitation by the author, so I have nothing to add.

I believe that the paper is logical and objective from the introduction to the conclusion. I believe that the scientific value of this study is equivalent to that of PLOS ONE.

Reviewer #2: Thank you for the opportunity to review this manuscript entitled “The value of D-dimer and platelet-lymphocyte ratio (PLR) combined with CT signs for predicting intestinal ischemia in patients with bowel obstruction”. The authors retrospectively analyzed the clinical and imaging date of patients diagnosed as bowel obstruction.

They concluded that the combined use of D-dimer, PLR and CT signs has high diagnostic value for intestinal ischemia in patients with bowel obstruction.

Although the manuscript are well written, I cannot accept the manuscript as the following reasons;

Major comments:

1. The inclusion criteria for this study should be more clearly described. This study focuses on cases that resulted in surgery as a result of bowel obstruction, but all cases that did not result in surgery seem to have no evidence of ischemia. It should be discussed how this point should be interpreted.

2. In this study, authors simultaneously perform multivariate analysis of categorical and continuous variables. It is unclear how the cut off value for continuous variables was set.

3. Were all cases registered in this study emergency surgical cases, or were conservative cases with no improvement also included? If cases without improvement by conservative treatment were registered, authors should consider timing from onset of symptoms to surgery as a factor in ischemia.

6. PLOS authors have the option to publish the peer review history of their article (what does this mean?). If published, this will include your full peer review and any attached files.

Reviewer #1: No

Reviewer #2: No

---

## [Author Response · Author response to Decision Letter 0]

27 Apr 2024

We wish to thank you for the time and effort you have spent reviewing our paper. We are pleased to note that you have found our research work interesting and also pointed out some problems to help us improve the quality of our study. Relevant suggestions and questions about the article have been corrected and answered in time.

---

## [Editor Report · Decision Letter 1]

30 Apr 2024

PONE-D-24-07576R1The value of D-dimer and platelet-lymphocyte ratio combined with CT signs for predicting intestinal ischemia in patients with bowel obstructionPLOS ONE

Dear Dr. Zhang,

Thank you for submitting your manuscript to PLOS ONE. After careful consideration, we feel that it has merit but does not fully meet PLOS ONE’s publication criteria as it currently stands. Therefore, we invite you to submit a revised version of the manuscript that addresses the points raised during the review process.

We look forward to receiving your revised manuscript.

Kind regards,

Yoshihisa Tsuji

Academic Editor

PLOS ONE

Journal Requirements:

Additional Editor Comments:

I am generally satisfied with the authors' responses to the Reviewer. So, please reflect more on the content of these responses in the main text. For example, regarding Q-1, please reflect more in the Discussion part. Q-2: Please provide details of how you determined the Cut-Off in the method section of the text more clearly. What Chinese sentences are sometimes included in the text? Please delete them if they are unnecessary.

---

## [Author Response · Author response to Decision Letter 1]

1 May 2024

We have made revisions based on your suggestions and hope that you will review our paper again.

---

## [Editor Report · Decision Letter 2]

27 May 2024

The value of D-dimer and platelet-lymphocyte ratio combined with CT signs for predicting intestinal ischemia in patients with bowel obstruction

PONE-D-24-07576R2

Dear Dr. Zhang,

We’re pleased to inform you that your manuscript has been judged scientifically suitable for publication and will be formally accepted for publication once it meets all outstanding technical requirements.

Kind regards,

Yoshihisa Tsuji

Academic Editor

PLOS ONE
---

## [Editor Report · Acceptance letter]

3 Jun 2024

PONE-D-24-07576R2 

PLOS ONE

Dear Dr. Zhang, 

I'm pleased to inform you that your manuscript has been deemed suitable for publication in PLOS ONE. Congratulations! Your manuscript is now being handed over to our production team.

Kind regards, 

on behalf of

Professor Yoshihisa Tsuji 

Academic Editor

PLOS ONE